# Ambient noise differential adjoint tomography reveals fluid-bearing rocks near active faults in Los Angeles

Xin Liu [1,2,3] ✉, Gregory C. Beroza [2] ✉ & Hongyi Li[3,4]

Water scarcity is a pressing issue in California. We develop ambient noise differential adjoint tomography that improves the sensitivity to fluid-bearing rocks by canceling bias caused by noise sources. Here we image the shallow S-wave velocity structure using this method beneath a linear seismic array (LASSIE) in Los Angeles Basin, which shows significant velocity reduction marking a major regional water producer, the Silverado aquifer, along with other fluid-bearing structures. Based on the S-wave tomography and previous P-wave studies, we derive the porosity in Long Beach and discover that the rock from 1-2 km depth surrounding the Newport-Inglewood Fault contains abundant fluids with pore-fluid fraction ~0.33. The high-porosity rock around the fault coincides with previously observed week-long shallow seismicity south of LASSIE array in Long Beach. The imaged S-wave velocity in the top layer shows a similar trend in the geotechnical layer Vs 30, suggesting additional applications to ground motion prediction.

The Los Angeles Basin was formed at the plate boundary of Pacific and North American plates. It hosts several active faults (Fig. 1), among which the most prominent are several NW-SE trending faults: the Newport-Inglewood Fault, Whittier Fault, and three blind faults – the Los Alamitos, Norwalk, and Lower Elysian Park Thrust Faults[1,2]. The strike-slip Newport-Inglewood Fault produced an M 6.3 earthquake in 1933, and still generates active seismicity as observed by nodal seismic arrays[3–5]. The Los Alamitos Fault is also seismically active but the recorded seismicity is weaker[6].

The near-surface structure in urban areas is important in several aspects. It supports the weight of building foundations, and the near-surface velocity/attenuation structure is important for ground motion prediction of potential earthquakes. It also hosts aquifers that could play an important role in the water supply, and the spatial distribution of the aquifers could also affect the subsidence of construction or liquefaction in times of an earthquake. Traditional methods for exploring the near-surface structure include seismic surveys with artificial sources (explosion or vibroseis), and electrical resistivity. The

former method is quite expensive and logistically difficult in an urban area, but can provide information on shallow water flowpath based on P-wave velocity[7]. The latter approach is more sensitive to the very shallow sediments in the top 50 m depth and has limited spatial coverage of ~300 m.

The Quaternary groundwater system in Long Beach has been studied using wells[8,9], which reach a maximum depth of 462 m. The San Pedro Formation of Lower Pleistocene age contains the most important groundwater supply among these: the Silverado aquifer. In particular, the borehole at Long Beach City College reveals the depth range of Silverado aquifer between 0.2–0.4 km. Deeper aquifers below the well depth (462 m) are not reported. The correlation between shallow seismicity and porosity around Newport-Inglewood Fault or Los Alamitos Fault in Long Beach has not been studied previously.

In this study, we develop ambient noise differential adjoint tomography to infer the near-surface shear-wave velocity structure beneath the linear LASSIE array in the Los Angeles Basin (Fig. 1). The

---

[1]Department of Earth Sciences, University of Hong Kong, Pokfulam, Hong Kong SAR, China. [2]Department of Geophysics, Stanford University, Stanford 94305, USA. [3]Key Laboratory of Intraplate Volcanoes and Earthquakes (China University of Geosciences, Beijing), Ministry of Education, Beijing 100083, China. [4]School of Geophysics and Information Technology, China University of Geosciences, Beijing 100083, China. ✉e-mail: liuxine@hku.hk; beroza@stanford.edu

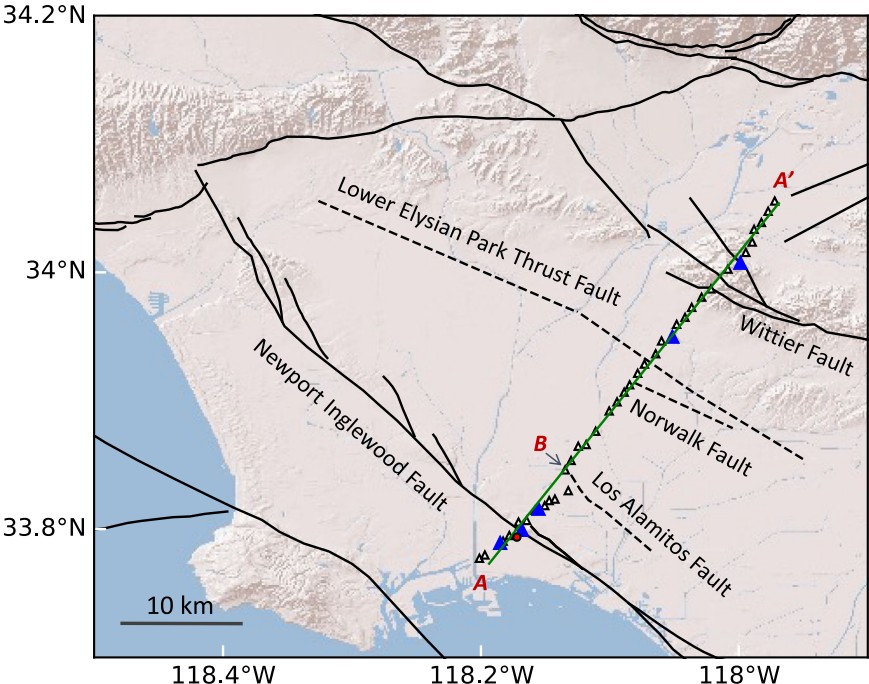

**Fig. 1 | Map of the LASSIE array and regional faults.** The red dot is the location of the LBCC-drilled well. The 5 blue triangles are the virtual sources for the validation dataset. The open triangles are the virtual sources for the training dataset. Black lines: faults (dashed if blind). A–A': the entire LASSIE array profile. A–B: the first 10-km LASSIE array profile for porosity estimation in Fig. 3. The topography base map was created using Esri "World Topographic Map" online API. https://basemaps.arcgis.com/arcgis/rest/services/World_Basemap_v2/VectorTileServer (8 September 2023).

LASSIE array consists of 42 broadband seismic stations that record the seismic noise data for one month in 2014. Previous studies of shear velocity structure using the LASSIE array[10,11] are mainly based on the ray-theoretical approaches, in which a phase velocity profile along the array is first created and interpreted through a 1D shear velocity inversion with depth for a layer-cake model at each horizontal grid location[12]. Our differential adjoint tomography successfully images a shallow aquifer between 0.2–0.4 km depth and other deeper fluid-bearing structures in the top 2 km depth due to the enhanced sensitivity to S-wave velocity structure. It has advantages over the traditional ambient noise ray-theoretical or adjoint tomography approach: (1) it is insensitive to the unknown distribution of noise sources, and (2) it incorporates the wave equation and finite-frequency effects that are not limited by the flat-layer assumption used in 1D depth inversion. In addition, to address the overfitting issue, we generate synthetic noise interferometry waveforms using the adjoint tomography result and minimize the observed and synthetic data misfit on the independent validation dataset of 5 virtual sources, which is common practice in training deep-learning neural networks. We infer the porosity value or its lower bound at/below 0.8 km depth based on the shear-wave velocity derived by ambient noise differential adjoint tomography and the existing P-wave velocity model. The results reveal abundant shallow pore fluids distributed in the known aquifers (e.g., Silverado aquifer between 0.2 and 0.4 km depth) and other deeper fluid reservoirs. One large fluid reservoir (between 1–2 km depth; 2 km width) with an average pore-fluid fraction of ~0.33 spans the Newport-Inglewood Fault asymmetrically to the northeast, and could reduce the normal stress acting on the fault plane and trigger the observed seismicity[4,5] near the fault in the upper 5 km. A comparison of the estimated shear-wave velocity in the top 100 m with previous Vs 30 values from geotechnical layers[13] shows surprising agreement in the general trends at the boundaries of geological units, suggesting future applications of the new differential adjoint tomography technique in geotechnical engineering.

## Results

### Love wave differential time adjoint tomography

We apply ambient noise differential time adjoint tomography to two different frequency bands: (1) low-frequency band 0.15−0.35 Hz, and 2) low+high-frequency band 0.15−1.5 Hz. Compared with the ray theory-based initial shear velocity model (Fig. 2A), the low-frequency result (Fig. 2B) shows significant velocity reduction, with up to 50% velocity decrease at 300 m depth and 23% decrease at 2 km depth southwest of the Newport-Inglewood Fault. In addition, there are several shallow zones of substantial velocity reduction up to 30% located in between the local faults northeast of the Newport-Inglewood Fault, suggesting that the fault planes act as bi-material interfaces[14,15] that separate geological units of different physical properties. The peak velocity reduction values for different shallow zones appear at ~300 m depth, suggesting sensitivity to layered structures at this depth. A previous study using Long Beach array also finds lower S velocity perturbation between 0.1–0.4 km depth near the southwest end of the LASSIE array[16], but their velocities at this depth range are ~35% faster than the results shown here at 300-m depth.

Combining the low and high-frequency differential time measurements, we obtain the final shear velocity model (Fig. 2C) with an enhanced image at shallow depths of 0–800 m. The final velocity model shows only a marginal difference from the velocity model based on low-frequency Love wave data, suggesting that the low-frequency data are also sensitive to the shallow structure in the top 800 m in addition to the deeper structure between 800−3000 m depth. We do not interpret the region for X between 26 and 35 km due to the significant topography of ~300 m as our program does not handle topography.

### Groundwater reservoir at 0.3 km depth in Long Beach

To investigate the hydrological properties of the low-velocity layers, we first estimate the Vp/Vs ratio based on the adjoint shear velocity tomography result. The P-wave velocity model is derived from the

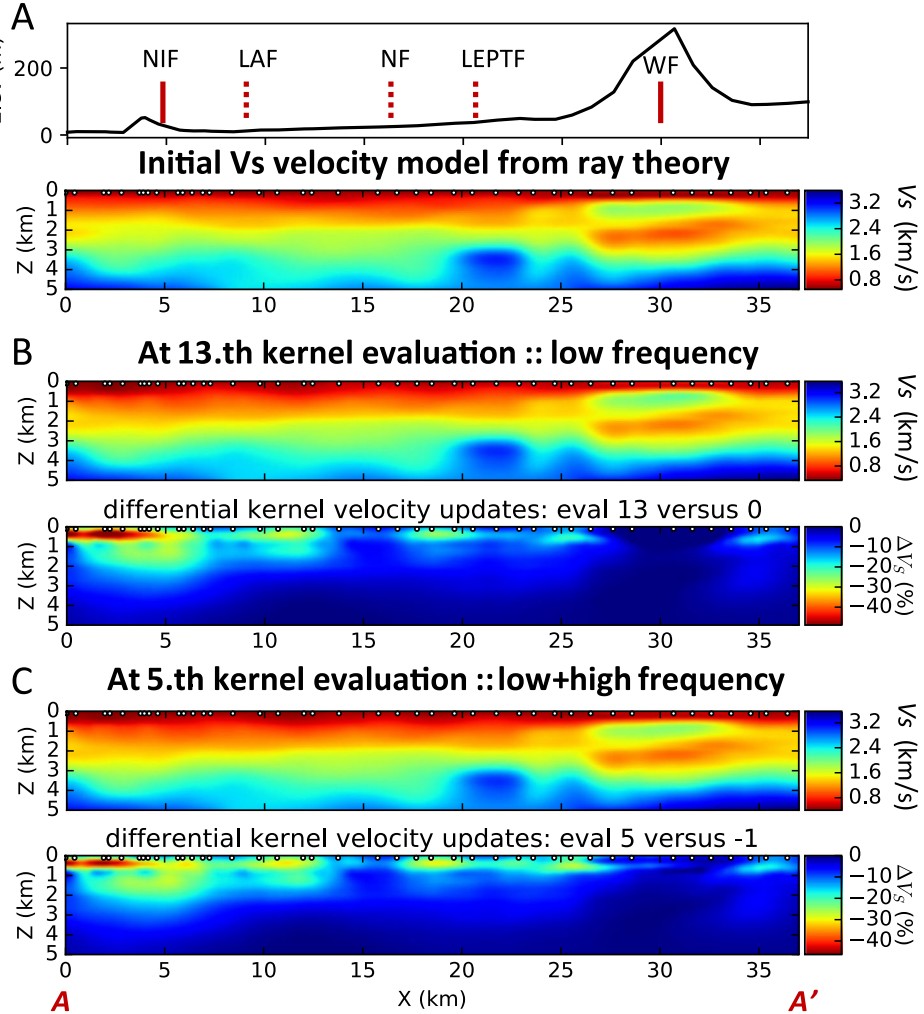

**Fig. 2 | S (Shear) wave tomography results. A** Topography along the LASSIE array with labels of fault locations (top). NIF Newport-Inglewood Fault, LAF Los Alamitos Fault, NF Norwalk Fault, LEPTF Lower Elysian Park Thrust Fault. Initial shear-wave velocity model based on ray theory (bottom). **B** Shear-wave velocity based on differential adjoint tomography and low-frequency data (top). Velocity update for the top panel compared with the initial ray theory model (bottom). **C** Shear-wave velocity based on differential adjoint tomography and low+high-frequency data (top). Velocity update for the top panel compared with the initial ray theory model (bottom).

Long Beach nodal array using ambient seismic noise interferometry[17]. For the overlapping part of the LASSIE and Long Beach nodal arrays, we compute the Vp/Vs ratios for the top 2 km of the sedimentary basin structure centered on the Newport-Inglewood Fault (Fig. 3a). Apart from the shallow layer in the top 100 m, the Vp/Vs ratio reaches a peak value of 4.5 at $X = 2$ km and 0.2–0.4 km depth southwest of the Newport-Inglewood Fault, which correlates with a known aquifer—the Silverado aquifer according to the sequence stratigraphy from a well at Long Beach City College[8]. Such large values are not unheard of, for example, similar and higher Vp/Vs ratios are found at Groningen, Netherlands in the top 0.2 km depth based on borehole seismic arrays[18]. Previous surface wave adjoint tomography with long-period data using the regional broadband seismic network[19] did not resolve the shallow layers in Los Angeles Basin where we find a high Vp/Vs ratio.

Across the Newport-Inglewood Fault, there is a 210-m uplift of the eastern block according to the structure map beneath Long Beach near Signal Hill[20], which probably brought the water-bearing structure to a shallower depth. Therefore, we cannot identify the Silverado aquifer between 0.2–0.4 km depth in the eastern block.

The porosity of rocks is the ratio of the volume of void space over the total volume of the rock. One theoretical framework could estimate directly the porosity information using adjoint state method[21], but it has no practical application to date. To convert the Vp/Vs ratio to porosity for water-saturated sediments, we adopt the Biot-Gassmann Theory by Lee (BGTL) method[22,23], which models the pore-fluid contribution to the Vp/Vs ratio for known rock frame properties. The model parameters include elastic moduli of both quartz and clay matrix materials (Supplementary Text S2). The BGTL method only predicts the porosity accurately for a differential pressure -15 MPa, which translates to a depth of 0.8 km, assuming an average density of 1.9 g/cm³ for the shallow sediment layers according to the SCEC community velocity model CVM-H[19]. Because the differential pressure is inversely correlated with Vp/Vs ratio, this method will overestimate the porosity above 0.8 km depth and underestimate the porosity below that depth.

For the same Vp/Vs ratio, the BGTL method will predict a much lower porosity for clay due to its much smaller shear modulus compared with sand. Therefore, the predicted porosity is too low for clay and too high for sand, suggesting that either clay or sand alone cannot explain the observed Vp/Vs ratio. Assuming the rock at shallow depth consists of 50% clay volume and 50% quartz, the estimated porosity at 0.8 km depth shows strong fluctuation between 0.25–0.38 (Fig. 3b).

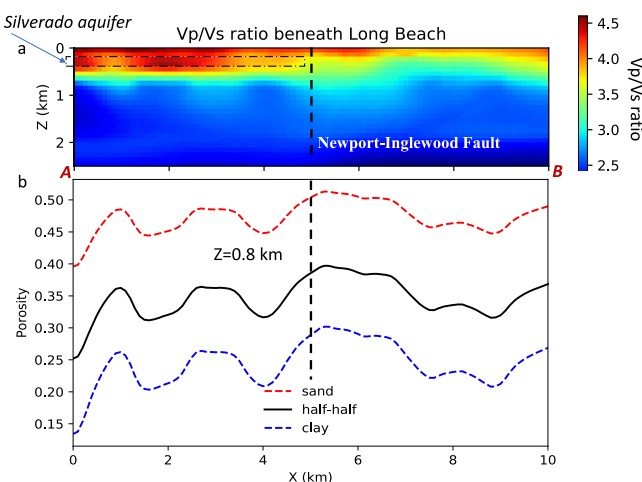

**Fig. 3 | Velocity ratio and porosity estimation. a** P and S-wave velocity ratio in the Long Beach section. **b** Inferred porosity at 0.8 km depth converted from Vp/Vs ratio for pure sand (red dashed line), pure clay (blue dashed line), and half sand-half clay (black solid line), respectively. The dashed vertical black line represents the Newport-Inglewood Fault.

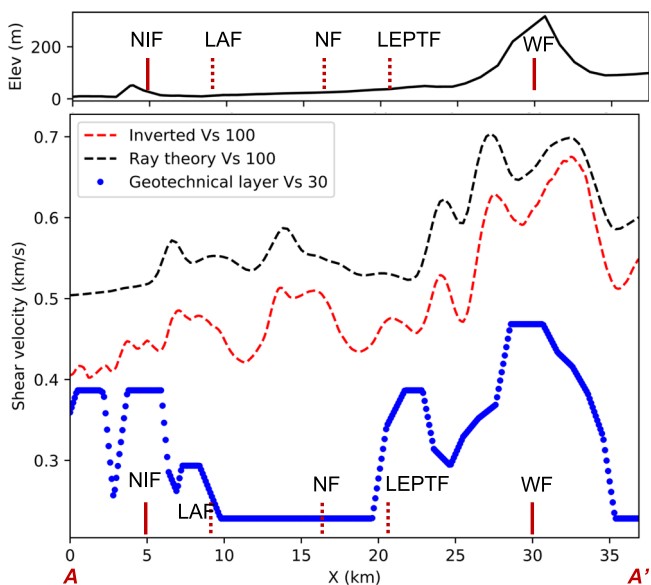

**Fig. 4 | Topography along the LASSIE array with labels of fault locations.** The fault acronyms are the same as in Fig. 2A (top). Comparison between inverted shear-wave velocity in the top 100 m and geotechnical layer Vs 30[13] (bottom).

At $X = 2$ km, the porosity is 0.31 at 0.8 km depth, which is a local minimum. In contrast, the Vp/Vs ratio peaks at the same horizontal location between 0.2–0.4 km depth, indicating greater pore-fluid fraction compared with other horizontal locations. Permeability is a measure of how easily water can move through the rock material. For sand, higher permeability correlates with higher porosity. For clay, its permeability is low despite higher porosity. The observations of lateral variations in porosity suggest that the sedimentary layer at 0.7-0.8 km depth is characterized by variable permeability (e.g. clay with different thickness), and the portion of the layer at X = 2 km is less permeable than nearby segments, thereby storing more pore fluids at shallow depths between 0.2–0.6 km while reducing the fluid below 0.8 km depth.

The two blocks separated by the Newport-Inglewood Fault show significant contrast in Vp/Vs ratio and porosity (Fig. 3), suggesting different materials for sedimentary layers across the fault interface. The sediments above 0.6 km depth in the NE block show lower Vp/Vs ratios than those from the SW block at the same depth range, indicating more pore fluids in the SW block at these shallow depths. For the sedimentary rocks situated below 0.7 km depth, the NE block shows higher Vp/Vs ratios within 2 km from the fault interface than the SW block, suggesting higher porosity for the NE block at greater depths near the fault interface. This observation is clarified by converting the Vp/Vs ratio beneath 0.8 km depth to a lower bound on porosity (Supplementary Fig. S1), which shows the deep reservoirs have porosity around 0.33.

### Near-surface structure and Vs 30

We compare the estimated shallow shear-wave velocity structure with the Vs 30 data from the geotechnical layer[13] beneath the LASSIE array (Fig. 4). The inverted shear-wave velocity is derived from the top 100 m due to the limitation of the grid size in adjoint tomography. The geotechnical layer was sparsely sampled (by strong motion stations) and Vs 30 values are grouped by different geological units[13].

The inverted shear-wave velocity in top 100 m (Vs 100) from adjoint tomography is considerably lower than that from ray theory tomography (Fig. 4). The trend of Vs 100 generally increases as the distance from the coast increases. For adjoint tomography, the Vs 100 southwest of the Newport-Inglewood Fault is the lowest, which is close the Vs 30 values from the geotechnical layer. For the block NE of Newport-Inglewood Fault, Vs 100 is higher while Vs 30 shows decreasing velocity.

For the geological unit between the Los Alamitos Fault and the Lower Elysian Park Thrust Fault, both the inverted Vs 100 (adjoint tomography) and the Vs 30 decrease at the edges of the geologic unit defined in the Vs 30 model. Vs 30 remains constant within the geological unit—perhaps due to sparse sampling, while Vs 100 exhibits lateral variations within the unit and is almost twice the value of Vs 30.

The high topography above the Whittier Fault corresponds to higher Vs 100 and Vs 30, while the inverted Vs 100 is ~0.15 km/s faster than Vs 30. In this case, the inverted Vs 100 from adjoint tomography matches the trend of the ray theory Vs 100, but is slightly lower.

## Discussion

We introduce ambient noise differential adjoint tomography to a linear array across the Los Angeles Basin. The inverted shear-wave velocity model based on differential time measurements of Love wave phases reveals low-velocity zones (up to 50% reduction compared with the ray theory velocity model) corresponding to groundwater/fluid reservoirs from 0.2 to 2 km depth where some deeper ones were not previously identified due to the limitation of borehole depth. Additionally, the inverted shear-wave velocity model in the top 100 m shows systematic agreements with the Vs 30 information from the independently derived geotechnical layer. Together, these results demonstrate the resolving power of the new differential adjoint tomography method compared with traditional ray theory tomography.

The porosity information is directly related to Vp/Vs ratios, but the P-wave velocity model is only available in the top 2 km and spans 10 km away from the Newport-Inglewood Fault due to the limitation of the Long Beach array[17]. Therefore, our study of porosity is limited to the smaller volume where both Vp and Vs models overlap. Another limitation is that the BGTL method only works for a depth range where the differential pressure is ~15 MPa. The P-wave velocity beneath Long Beach (Supplementary Fig S2), however, contains much less lateral variation than the S-wave velocity. Therefore, most of the observed lateral variations in Vp/Vs ratio arise from the S-wave velocity model, which is more sensitive to pore fluids in sedimentary rocks than the P-wave model. Assuming the differential pressure is similar at the same depth, the first limitation can be alleviated by comparing the lateral variation of shear velocity update after adjoint tomography (e.g., Fig. 2B, C), as the shear velocity reduction indicates larger Vp/Vs ratio

and therefore higher porosity. The second limitation can be reduced by estimating the lower bound of porosity for deeper depths with higher differential pressure (e.g., Supplementary Fig S1).

The spatial distribution of porosity and its lower bound have important implications for hydrological studies in the Long Beach sedimentary basin. The higher porosity directly correlates with greater S-wave velocity reduction, which allows us to use the S-wave velocity model anomaly (Fig. 2C) to predict relative fluid abundance at the same depth for a broader cross-section from Long Beach to Anaheim (X from 0 to 25 km). For example, the area to the east of LAF (X = 10-13 km; depth between 0.2-1.8 km) also corresponds to significant Vs reduction of 20-30%. The shallow sections (0.1-0.5 km depth) between NF and LEPTF or at the western foothills (X = 25 km) also feature significant Vs reduction 20-25%. These results suggest that the aquifers at 0.2–0.4 km depth have spatially variable pore fluids, and the water-bearing reservoirs at different depths can be interconnected by vertical channels with higher permeability than surrounding rocks. The abundant fluids around the Newport-Inglewood fault suggest vertical migration of water from surface to aquifers at different depths, which provides a means of fluid replenishment for the Silverado aquifer. Based on the spatial variation of pore fluids derived from tomography, future studies of both horizontal and vertical fluid migration patterns can help achieve sustainable groundwater extraction and avoid drilling unnecessary water wells in general, and in Los Angeles or other large cities in particular. Ambient noise differential adjoint tomography is therefore a cost-effective way of identifying aquifers and locations with higher fluid content, optimizing the drilling of new water wells, and helping to provide access to freshwater for big cities.

Moreover, the existence of abundant pore fluids around the Newport-Inglewood Fault coincides with the higher seismicity in the top 5 km on the same fault observed by the Long Beach phase B array to the south of the LASSIE array[4,5], possibly because the higher porosity reduces the effective normal stress on the fault plane, increasing the microseismicity. Although the fluid within the Silverado aquifer flows easily and cannot increase the pore pressure (e.g., Townend and Zoback, 2000), the deeper fluid reservoir between 0.8–2.2 km depth may be sealed and can increase the pore-fluid pressure due to the combination of hydrocarbon reservoir beneath Signal Hill and water injection. This is analogous to induced earthquakes[24] and perhaps some natural earthquakes in southern California[25] being induced by fluid pressure variations.

The estimated shallow shear-wave velocity (Vs 100) also shows a similar trend to the geotechnical layer Vs 30. Admittedly, the geotechnical layer does not have good resolution because it assumes similar velocity within each geological unit and is sparsely sampled[13]. The adjoint tomography result clearly shows gradients of velocity at the boundaries of geological units, and the internal velocity variation within each geological unit. One drawback of this analysis is that the inverted Vs 100 represents the average shear velocity in the top 100 m as opposed to top 30 m in geotechnical layer Vs 30. For future work, it would be possible to increase the shallow depth sensitivity by densifying the seismic array and pushing the high-frequency limits. Despite the difference in depth sensitivity, our method shows potential in future high-resolution and low-cost geotechnical applications.

Ambient noise differential adjoint tomography shows great potential for resolving detailed geological/hydrological structures in urban sedimentary basins. It reduces the bias caused by uneven and temporally variable noise source distribution. Moreover, to address the overfitting problem in the optimization step, we use 5 stations as virtual sources for a validation dataset, which is independent of the training dataset consisting of 36 virtual sources. By minimizing the validation error, we choose the optimal velocity model that do not overfit to the training dataset. This approach is widely used in machine learning and could be useful for adjoint tomography community. It is similar to cross-validation, but the multiple different combinations of training and validation sets required for true cross-validation are not feasible due to the computational demands of adjoint tomography.

Some previous studies have overlapping areas with the LASSIE array. Using the regional broadband network in Southern California, a more recent study[26] combines the phase velocity and ellipticity information from ambient noise Rayleigh wave to estimate the near-surface S = wave velocity model, which improves upon the SCEC community velocity model. Another study derives the S velocity model with the Long Beach nodal array[16], which contains more than 5200 receivers with average spacing of 0.1 km, while the LASSIE array has an average spacing of 1 km. They use Eikonal tomography to estimate phase velocity map at different frequencies, which takes the spatial gradient of travel time for each virtual source. Unlike the adjoint tomography which directly estimates S velocity, the Eikonal tomography converts phase velocity maps to S velocity structure through 1D depth inversion below each geographic location.

The Vp and Vs models are derived from different seismic arrays with different approaches. The Vp model is based on P-wave phases extracted from the noise interferometry data of the 2D Long Beach nodal array through ray tracing of refracted P-wave ray paths[17]. The P-wave ray path between a virtual source and a receiver turns at approximately 2 km depth and becomes almost vertical near the surface, which is not sensitive to the shallow, thin aquifer system due to the nearly vertical incidence. The Vs model derived here is from the ambient noise differential adjoint tomography based on wave equations and a linear array, which involves surface waves traveling horizontally and is suitable for resolving lateral variations of Vs in the shallow aquifer system. Therefore, it's possible that some unresolved P-wave velocity variations may be missing in the Vp/Vs ratio (Fig. 3a). Due to the density of Long Beach nodal array and the proximity to the coast, however, the unresolved Vp anomaly at the shallow depths (top 1 km) is unlikely to be as significant as the Vs anomaly. For future work, it would be interesting to implement double-difference adjoint tomography[27] for the same P-wave travel time data from Long Beach array, which can potentially improve the P-wave velocity tomography result.

## Methods

A major issue in ambient noise adjoint tomography is that the noise interferometry function depends on both the noise sources and the structural properties. Given the complex nature of the earth's heterogeneous ambient seismic field at various frequency bands and its temporal variation, the Green's function derived from the noise interferometry function is inevitably biased by the noise sources. The solution we propose is to use differential time measurements of two station pairs within a linear triplet of stations, such that the differential time is a small fraction of the travel time between the longest station pair within the triplet[11,28].

Double-difference adjoint tomography using P waves has been applied to northeast Japan[29] and Alaska[27] for crust and upper mantle structure. For earthquake body wave data, this method is capable of canceling the signature of earthquake source radiation pattern, which is helpful in enhancing the resolution of the tomographic image.

The ambient noise differential adjoint tomography has a similar goal of suppressing the noise source signature. Suppose there are three stations on the same line, numbered consecutively (Fig. 5A). Station 1 is the virtual source in seismic interferometry. Station pair 1−3 has slightly longer distance than station pair 1−2. To ensure maximum overlap of the Fresnel zones of these two station pairs, we choose the station pair 1−3 such that its distance is less than 125% of that of pair 1-2. The angle between 1−2 and 1−3 station pairs should be less than 5 degrees. By taking the differential time measurements for the noise interferometric functions from pairs 1−3 and 1−2, the source kernel sensitivity within the overlapping area is effectively canceled, creating

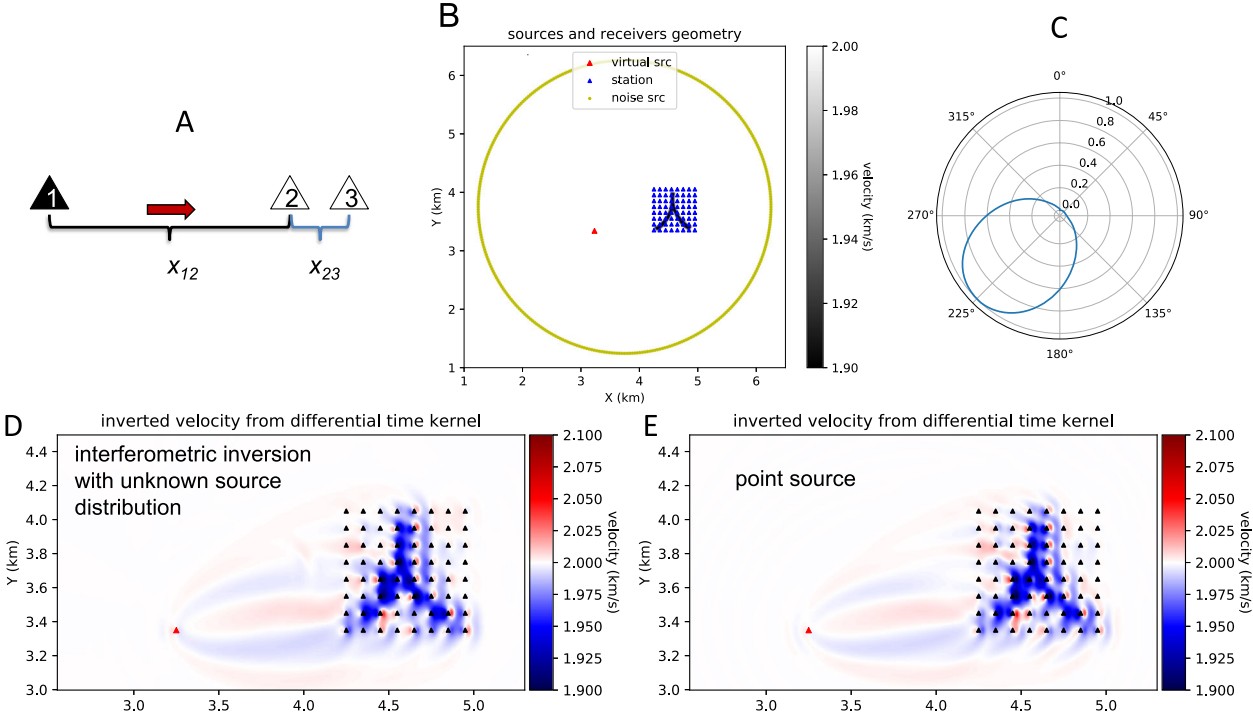

**Fig. 5 | Ambient noise differential adjoint tomography. A** Differential time measurements for a typical station triplet: one virtual source (1) and a pair of receivers (2, 3). **B** Geometry for the forward simulation: one virtual source (red) and a subarray (blue). **C** Noise source intensity distribution versus azimuth for the synthetic noise interferometry data. **D** Inverted velocity based on interferometric differential time kernel. **E** Estimated velocity based on point source differential time kernel.

a misfit function that is more sensitive to structural properties between stations 2–3[28,30].

## Replacing the virtual source by a point source
Previous work[28] demonstrated that differential time sensitivity kernels of noise interferometry functions are useful for the general case when the heterogeneous distribution of noise sources is unknown. Due to a weak sensitivity to noise sources, the differential time kernel for unknown noise source distribution can recover the velocity anomaly nearly as well as the case in which there is complete information on the noise source distribution. Moreover, because the point source kernel is equivalent to the noise interferometry kernel for uniform noise sources, we can make further simplifications by replacing the virtual source with a point source while still using the differential time measurements. This simplification extends the ambient noise differential time kernels to linear arrays where 2D station coverage is not available.

We illustrate this approximation with synthetic tests. The geometry of forward simulation contains a virtual source and a subarray of $8 \times 8$ stations, enclosed by a ring of noise sources (Fig. 5B). The observed noise interferometry functions are generated between the virtual source and each of the stations in the subarray following Liu, 2020. The lambda-shaped low-velocity structure is within the subarray. The azimuthal noise source intensity distribution is maximum in SW direction and zero in NE (Fig. 5C).

Based on the simulated data, we estimate the velocity anomaly structure with two approaches: 1) assuming unknown noise source distribution, we use uniform noise source distribution with interferometric differential time kernels[28] to perform adjoint tomography (Fig. 5D); 2) replacing the virtual source by a point source, we perform differential time adjoint tomography (Fig. 5E) with point source kernels[31]. The adjoint tomography results for both cases are nearly identical with minor differences in the side lobes, suggesting that the interferometric kernel for uniform noise sources is equivalent to point source kernel for differential time kernels subject to the distance and direction criteria for linear station triplets in this study.

## Differential time kernels for a point source
Here we briefly review the key formulae for differential time kernels for a point source. For non-dispersive signal, we first define the misfit function for differential time measurements,

$$\chi_{23}^{dd} = \frac{1}{2}\left[\Delta t_{23} - \Delta t_{23}^{obs}\right]^2,\qquad(1)$$

The variation of the misfit function is expressed as an integral over the sensitivity kernels,

$$\delta\chi_{23}^{dd} = \int_\Omega \left[K_C^{dd}(\mathbf{x})\frac{\delta C}{C} + K_\rho^{dd}(\mathbf{x})\frac{\delta\rho}{\rho}\right]d\mathbf{x},\qquad(2)$$

where $K_C^{dd}$ is the sensitivity kernel for the elastic tensor $C(\mathbf{x})$, $K_\rho^{dd}$ is the sensitivity kernel for density.

The differential time sensitivity kernel is the interaction between the forward wavefield $\mathbf{u}(\mathbf{x})$ and the adjoint wavefield $\mathbf{u}^\dagger(\mathbf{x})$,

$$K_C^{dd}(\mathbf{x}) = -C(\mathbf{x})\int_f \nabla\mathbf{u}(\mathbf{x},\omega)\left[\nabla\mathbf{u}_3^\dagger(\mathbf{x},\omega) - \nabla\mathbf{u}_2^\dagger(\mathbf{x},\omega)\right]d\omega,\qquad(3)$$

where $\mathbf{u}_3^\dagger$ and $\mathbf{u}_2^\dagger$ are two adjoint wavefields from locations $\mathbf{x}_3$ and $\mathbf{x}_2$, respectively.

For SH waves (e.g., Love waves) propagating in a 2D cross-section along a line, the above vector wavefield $\mathbf{u}(\mathbf{x})$ reduces to displacement in crossline direction, $u_y$. Suppose the medium is isotropic, we

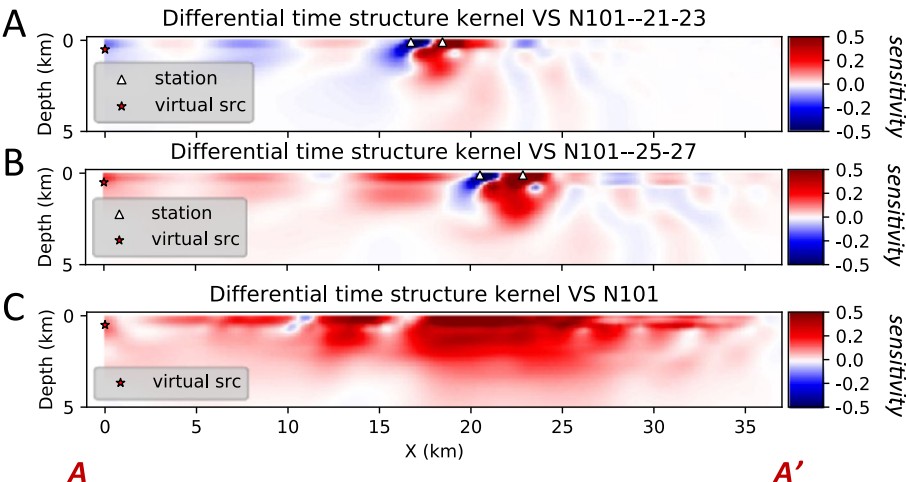

**Fig. 6 | Differential time structure sensitivity kernels for different station triplets.** The sensitivity kernel is normalized between −1 and 1. **A** Structure kernel for virtual source N101 and receiver pair 21–23. Station N101 is in the SW corner of the LASSIE array. Stations are numbered consecutively from SW corner. **B** Structure kernel for virtual source N101 and receiver pair 25–27. **C** Combined sensitivity kernel for the virtual source N101 and all station triplets.

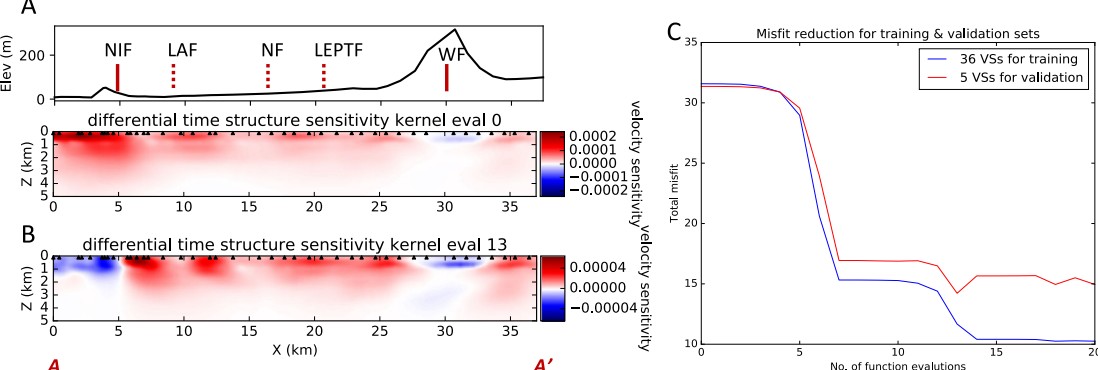

**Fig. 7 | Iterative inversion for the differential adjoint tomography using low-frequency data. A** Top: topography along the LASSIE array with fault locations. Bottom: total sensitivity for all 36 virtual sources in the training set at the initial evaluation of the kernel. **B** Total sensitivity for all 36 virtual sources at the 13th evaluation of the kernel, which uses the optimal velocity model that minimize the validation set. **C** Training and validation misfit (loss) functions versus iteration. The optimal velocity parameter is reached at the 13th iteration, where the validation misfit is minimum.

consider the differential sensitivity kernel for shear modulus μ,

$$K_\mu^{dd}(\mathbf{x}) = - \mu(\mathbf{x}) \int_f \nabla u_y(\mathbf{x}, \omega) \cdot \left[ \nabla u_{y3}^\dagger(\mathbf{x}, \omega) - \nabla u_{y2}^\dagger(\mathbf{x}, \omega) \right] d\omega, \quad (4)$$

where the · represents vector inner product.

For non-dispersive signals, the differential time adjoint sources corresponding to the adjoint wavefields $u_{y3}^\dagger$ and $u_{y2}^\dagger$ are, respectively[31–33],

$$g_{13}^\dagger(t) = [\Delta t_{23} - \Delta t_{23}^{obs}] \frac{\partial_t u_{y2}(T-[t+\Delta t_{23}])}{\int_0^T \partial_t^2 u_{y2}(t+\Delta t_{23}) u_{y3}(t) dt} \delta(\mathbf{x} - \mathbf{x}_3)$$

$$g_{12}^\dagger(t) = [\Delta t_{23} - \Delta t_{23}^{obs}] \frac{\partial_t u_{y3}(T-[t-\Delta t_{23}])}{\int_0^T \partial_t^2 u_{y2}(t+\Delta t_{23}) u_{y3}(t) dt} \delta(\mathbf{x} - \mathbf{x}_2)$$
$$(5)$$

where $u_{y2}(t)$ and $u_{y3}(t)$ are forward synthetic waveforms recorded at receiver locations $\mathbf{x}_2$ and $\mathbf{x}_3$, respectively.

**Differential time kernels for Love waves and inversion**
For dispersive seismic waves, the differential time adjoint sources are defined in Supplementary Text S1. We use finite-difference method to compute the elastic wavefield for forward and backward simulations.

The starting shear velocity model for adjoint tomography is the standard 1D layered shear velocity inversion for each X grid location along the linear array.

Based on eq. (S1), the differential time structure kernel for one triplet is primarily sensitive to the velocity structure between the two nearby receivers. For the virtual source N101, the station triplet containing the receiver pair 21-23 shows strong sensitivity between the two nearby receivers where differential time data are measured (Fig. 6A). Similarly, strong structure sensitivity also concentrates between the receiver pair 25-27 (Fig. 6B) for the corresponding station triplet with larger station spacing. For both station triplets, there are asymmetric side lobes to both sides of the differential time receiver pair. The blue side lobe next to the left receiver (closer to virtual source) is possibly caused by differential time measurements, finite-frequency effect and the Born approximation used in computing the Fréchet derivative of the misfit function, but the amplitudes of the side lobes are much smaller than the main lobe.

To sum the contributions from all triplets for one virtual source, it is more efficient to compute the combined sensitivity kernel using eq. (S10), which requires only one forward simulation and one adjoint simulation. The combined sensitivity kernel for the virtual source N101 (Fig. 6C) shows positive shear velocity sensitivity from near-surface to ~3.5 km depth, suggesting that the actual shear velocity is slower than

in the starting model. Moreover, the side lobes are significantly reduced due to the overlapping of multiple station triplets. When we combine all the virtual sources with possible station triplets, we can suppress most of these oscillations (Fig. 7A). The only remaining effect of side lobes would appear at the edges of an array, but the side lobes are still much weaker than the main lobes, otherwise the misfit functions (Fig. 7) for training and validation sets would not decrease.

For the adjoint tomography at long periods (0.16–0.35 Hz), we select 36 stations as virtual sources for structure kernel computation. The other 5 stations are used to prevent overfitting. At each iteration, we sum the kernels from different virtual sources directly as we sum their corresponding misfit values. At the initial iteration (Fig. 7A), the sensitivity kernel of 36 virtual sources shows strong positive sensitivity to low-velocity structures at shallow depths: down to 3.5 km SW of the Newport-Inglewood Fault; 2 km depth between Newport-Inglewood Fault and Lower Elysian Park Thrust Fault). At the 13th iteration (Fig. 7B), the absolute values of the sensitivity are considerably smaller than the initial iteration, and the positive and negative sensitivity values may correlate with some residual data misfit around fault traces.

In the iterative adjoint tomography, the total misfit decreases progressively as the number of iterations increases (Fig. 7C)[34]. The optimization converges around the 14th iteration. The validation step using the other five stations for virtual sources, however, suggests that the minimum validation misfit occurs at the 13th iteration, and the 14th iteration actually increases the validation misfit, suggesting overfitting of the observed data. Therefore, we choose the inverted shear velocity model at the 13th iteration as the adjoint tomography result for the long-period data.

## Data availability
All results needed to evaluate the conclusions in the paper are present in the paper and/or the Supplementary Materials. The S-wave adjoint tomography model is deposited on Zenodo: DOI:10.5281/zenodo.8376681. The quaternary fault lines are from the SCEC Community Fault model (https://www.scec.org/research/cfm) and the USGS quaternary fault database (https://doi.org/10.5066/F7S75FJM). The LASSIE array data (https://doi.org/10.7909/c3fx77k9) are available on the IRIS data service website (https://ds.iris.edu).

## Code availability
The wavefield simulations are generated using Madagascar (https://github.com/ahay/src).

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

## Acknowledgements

This work was supported by the junior faculty startup fund from the University of Hong Kong. H.L. and X.L. acknowledge the support from the National Natural Science Foundation of China (Grant no. U1939203). G.C.B. and X.L. acknowledge the support from Southern California Earthquake Center grant award no. 20208. X.L. acknowledges the Seed Fund (no. 2201101676) support from the University Research Committee of HKU.

## Author contributions

G.C.B. conceived the study and advised on the technical & scientific interpretation of results. X.L. developed the method of ambient noise differential adjoint tomography, analyzed the data, and interpreted the results. H.L. performed the synthetic test and evaluated data misfit. All authors contributed to the writing and revision of the manuscript.

## Competing interests

The first author X.L. declares the existence of competing interest in the form of a pending US Provisional patent application. Applicant: HKU (institute). Inventor: Xin Liu. Status: pending. Application no. 63/510,624. The aspects of the manuscript covered: the methods section and Supplementary Material Text. Other authors declare no competing interest.
