## [Peer Review File · Nature Communications]

REVIEWER COMMENTS

Reviewer #1 (Remarks to the Author):

SUMMARY

This is an impactful paper, in terms of methodological developments and also a societally relevant application. The method (ambient noise differential adjoint tomography) is a new variant on two methods that are relatively new to seismology. Ambient noise adjoint seismology was first performed by Min Chen et al. (2014 GRL: Low wave speed zones in the crust beneath SE Tibet revealed by ambient noise adjoint tomography), while travelttime differential adjoint tomography has been performed with P waves by J. Chen (2022) and P. Chen (2023). Here the authors blend these approaches and aim the technique at a compelling target: shallow, groundwater-bearing rocks in the Los Angeles region. (The image spans a linear profile and is therefore 2D.) To me, Figure 3 (top) is the key result, as it clearly shows a contrast in elastic properties across the fault; the authors make a compelling case that this is due to difference in rock properties and groundwater reserves. The authors have done due diligence in presenting their methods as well as their caveats. I have some minor suggestions that can likely be addressed with minor updates.

MAIN POINTS

+ There are some key recent papers whose work should be considered and possibly cited. I think that these two papers are the first applications of differential adjoint tomography:

Chen, J., Chen, G., Wu, H., Yao, J., & Tong, P. (2022). Adjoint tomography of northeast Japan revealed by common-source double-difference travel-time data. *Seismological Research Letters*, 93(3), 1835–1851. <https://doi.org/10.1785/0220210317>
<https://pubs.geoscienceworld.org/ssa/srl/article/93/3/1835/612501/Adjoint-Tomography-of-Northeast-Japan-Revealed-by>

Chen, G., Chen, J., Tape, C., Wu, H., & Tong, P. (2023). Double-difference adjoint tomography of the crust and uppermost mantle beneath Alaska. *Journal of Geophysical Research: Solid Earth*, 128, e2022JB025168. <https://doi.org/10.1029/2022JB025168>
<https://agupubs.onlinelibrary.wiley.com/doi/10.1029/2022JB025168>

The distinction is that they used P waves, whereas the current study uses ambient noise surface waves. But the fact that the principles seem to work in a real problem was demonstrated with P waves.

A paper using adjoint tomography (or "full waveform inversion") to examine groundwater reservoirs is this one:

Wang, W., Nyblade, A., Mount, G., Moon, S., Chen, P., Accardo, N., et al. (2021). 3D seismic anatomy of a watershed reveals climate-topography coupling that drives water flowpaths and bedrock weathering.

Journal of Geophysical Research: Earth Surface, 126, e2021JF006281.
<https://doi.org/10.1029/2021JF006281>
<https://agupubs.onlinelibrary.wiley.com/doi/full/10.1029/2021JF006281>

It's possible that the scales and techniques are quite different, but it may be worth citing.

+ The theory to estimate porosity directly (rather than indirectly, as done here) is fully established (Morency, Tromp, 2008, 2009, 2011) and not yet applied for 3D real problems, as far as I know.

I think that Andrea Revil also tackled poroelastic wave propagation and seismic imaging:
<https://scholar.google.com/citations?user=dsQqrkwAAAAJ&hl=en>

+ I wonder if the V_p/V_s contrast across the fault could be demonstrated more clearly by comparing a simple depth profile of V_p/V_s for the average of one side and the average of the other side. Potentially this could be included in Figure 3.

MISC POINTS

+ Put and A and A' in Figure 3 or just NE and SW at the ends of the top image.

+ I did not see any mention of what method was used to solve the 2D elastic wave equation (finite difference? spectral element?). This should be stated in the Methods.

+ Is topography taken into account? It is shown in Figure 2a, but is it accounted for in the forward wavefield simulations?

+ Is the numerical grid regular or are there smaller elements near the surface in order to efficiently accommodate the lower wave speeds? Certain considerations would be essential for extension to 3D.

+ Rayleigh wave ellipticity measurements, such as those used in Berg et al. (2018 JGR), have the potential to support very shallow seismic imaging. These can be extracted from either earthquake waves or ambient noise.

+ Castellanos2020 is cited for the V_p model. It seems like there is potential to use a similar approach to get V_p as was used to get V_s . For example, could the P waves from Castellanos2020 or from earthquakes be used within a differential adjoint tomography framework? This would ensure that the 3D wavefield simulations were directed at both V_s and V_p . The authors address this topic in the final paragraph of the paper, and I'm not suggesting that more needs to be stated.

Reviewer #2 (Remarks to the Author):

This study focuses on a new adjoint tomography method which improves existing imaging methodologies in two important ways: (1) produces better images of the shallow seismic structure using ambient noise, and (2) minimizes the bias from uneven noise source distribution which often

leads to inaccurate velocity estimations. The authors applied this method to a field case study, demonstrating its feasibility. Generally speaking, I think this study is worthy of publication, but would need a major revision of the physical connection between the seismic observations and their geophysical significance/importance to the LA Basin/NIF. I'd like to point out the following comments/questions:

Adjoint Method

1. although the authors have addressed the advantage of this new method compared to the conventional ray-theory inversion method (i.e. lines 51-58), it is still a little unclear to me how much increased sensitivity is expected. I assume this new method can do a better job to image the seismic velocity heterogeneity (because this method is able to identify a shallow aquifer which was not seen from other tomography studies). Better highlighting this would enhance the paper.
2. It might be good to highlight the importance of studying near-surface seismic structure. why should we care about the near-surface seismic structure, and what are the challenges in doing shallow structure imaging using the existing techniques?
3. It might be better to place one sentence to introduce what LASSIE array is (i.e., what kind of data is it recording? line 50)

Vp/Vs – Porosity Connection and Geophysical Interpretation

1. The derived porosity based on vp/vs is plausible. I wonder if there is any other evidence/data/reference to support the derived porosity value. Some follow-on questions that might help you answer the above question, but don't necessarily need to be addressed specifically – perhaps a geologic cross section exists along the same cross section of the NIF as Figure 3 and could show some geologic heterogeneity? Is the San Pedro Formation not present on the east side of the fault? How much throw is on the NIF? Is that coincident with your Vp/Vs gradient across the fault?
2. The words "porosity" and "permeability or permeable" should not be mixed (i.e., line 140). High porosity rock does not always lead to high permeability. Clays are an example of a rock type that exhibit high porosity and low permeability. I'd encourage the authors to mention the definition and address the correlation between the two in the manuscript.
3. The porosity predictions are quite sensitive to the assumed geology (i.e. clay vs. sand fraction). There is a difference of more than 100% in porosity. What doesn't quite make sense to me – clay's usually have much higher *porosities* than sands. Clay's typically have total porosities in the 40-50% range but very low effective porosity (1-2%) due to the plate-like arrangement of grains. This very low effective porosity gives clays their

extremely low permeability. Sands typically exhibit total porosities in the 20-25% range, but have orders of magnitude higher permeabilities as the spherical grains produce effective porosities near the total porosity. Is the BGTL model actually sensitive to effective porosity and not total porosity? Effective porosity is more directly related to permeability...

4. Line 220-221, the authors argue the seismicity around the Newport-Inglewood fault could be related to high amount of pore fluids suggested by high porosity (~0.33). Higher porosity values, in and of themselves, do not mean that fluid pressure is higher or lower in this portion of the aquifer. Porosity influences aquifer storage, but not necessarily fluid pressure. Higher porosity rocks tend to be higher permeability (with some exceptions like those noted above), but higher permeability would actually tend to indicate lower fluid pressures are hydraulic gradients would not build as easily. We expect the groundwater flow to occur more easily without causing high pore pressure to accumulate, so the fault can stay strong (e.g., Townend and Zoback, 2000).

Grammatical Edits

1. Line 43: lower Pleistocene age
2. Line 46: deeper aquifers? Water is likely present at deeper depths but perhaps not aquifer quality/quantity
3. Line 166: check for grammatical clarity
4. Line 182: check for grammatical clarity

Reviewer #3 (Remarks to the Author):

Summary:

The authors propose an ambient noise differential adjoint tomography method, apply it to the LASSIE array, and further come out with a shear wave velocity model beneath the LA basin. Owing to the advantages of using differential time information between receivers, they eliminate the potential bias caused by unknown non-uniform noise sources. More than a velocity model, they also estimate V_p/V_s ratio and porosity according to some previous works. They found extremely high V_p/V_s ratios at the place where the Silverado aquifer was observed and indicating the existing pore fluids.

Comments:

1. Data information needs to be included somewhere in the manuscript.
2. In the synthetic results (Figure 5), the main feature was revealed, but there are many oscillations at sidelobes. As well as Figure 6, many oscillations exist in the kernel profiles. What is the cause of these oscillations? Are these patches appear due to finite frequency bands? For the V_s models

(Figure 2), most variations are in shallow portions, and they look like the oscillations in the synthetic tests. Can the authors clarify these small patches of anomalies from the oscillations as the synthetics? The fluctuations appear in Figure 3 as well at ~ 1 km to the surface.

3. Line 122-124. This sentence may not be appropriate. The purpose and the scales of this study and the cited paper (Shaw et al., 2015) are different. Why do the authors want to mention Shaw et al. (2015) as a point? Moreover, there are many other Vs models that have been published (e.g., Lin et al., 2013; Berg et al., 2018, etc.) but have yet to be mentioned in the manuscripts. The authors may want to add paragraphs to demonstrate the comparison between their Vs model and other published Vs models.
4. What is the spatial resolution for the Vs model (this research) and the used Vp model (the referred article) in the Long Beach section? The nodal array for the Vp model (Castellanos et al., 2020) looks much denser than the LASSIE array, but the Vp profile (Figure S2) looks like a reference 1-D model. As mentioned in many places of the manuscript, Vp and Vs models are from different methods and have different sensitivity to the structures. It would be good to discuss how these two models are sensitive to the structures differently. As said in Lines 244-252, there may be existing some unresolved velocity anomalies.
5. To study the structures at Long Beach, there is a comparable Vs model using the same nodal array as the Vp model proposed by Lin et al. (2013). What is the resolution difference between the Vs model in this study and Lin's model?
6. In Figure 3 black curve (and Line 136), the local minimum porosity ($X \sim 1.6$ km) has a bit shift to $X = 2$ km (where the highest Vp/Vs ratio appears at the top plot). Can the authors clarify this?
7. The manuscript mainly discusses the Long Beach section's Vp/Vs ratios and porosity. How about the rest of the areas in the profiles? For instance, there is a velocity reversal at a distance between 25-35 km in Figure 2.
8. While the topography along the LASSIE array does not change much in the WS part, where the paper mainly focuses, do the authors perform/consider elevation correction in the resulting model since they target shallow top 1 km depth?
9. In the last part of the main text (Lines 356-380 and Fig7), the authors demonstrate how the iterative process goes throughout the iteration. The results shown in this section are based on the low-frequency band (0.16-0.35 Hz). How does the iterative process go when using low+high frequency signals (as shown in Figure 2C)?
10. Using a small dataset to validate the output model is common when resolving scientific problems. This concept is not originally from the machine learning or deep-learning community. It seems unnecessary to mention deep-learning neural networks (Lines 61-62) or machine learning (Lines 240-241).
11. What is the unit of misfit? The values of the misfit in Figure 7 look large.
12. It would be good to mention Fig 2C in the paragraph (Lines 100-105).
13. Lines 169-171: What do the authors mean that both Vs100 and Vs30 "decrease" at the edge of the unit? Consider marking the faults in the bottom plot of Figure 4.
14. How do the five virtual source stations be selected?

For figures:

- Figure 1: (1) The solid triangles and the red dot (the well location) are hard to read. (2) consider marking the section of Figure 3.
- Figure 4: Consider marking the faults in the bottom plots.

Adding color-code units (or components) on those figures with color bars would be better.

References

Castellanos, J. C., Clayton, R. W., & Juarez, A. (2020). Using a time-based subarray method to extract and invert noise-derived body waves at Long Beach, California. *Journal of Geophysical Research: Solid Earth*, *125*(5), e2019JB018855.

Shaw, J. H., Plesch, A., Tape, C., Suess, M. P., Jordan, T. H., Ely, G., ... & Munster, J. (2015). Unified structural representation of the southern California crust and upper mantle. *Earth and Planetary Science Letters*, *415*, 1-15.

Lin, F. C., Li, D., Clayton, R. W., & Hollis, D. (2013). High-resolution 3D shallow crustal structure in Long Beach, California: Application of ambient noise tomography on a dense seismic array. *Geophysics*, *78*(4), Q45-Q56.

Berg, E. M., Lin, F. C., Allam, A., Qiu, H., Shen, W., & Ben-Zion, Y. (2018). Tomography of Southern California via Bayesian joint inversion of Rayleigh wave ellipticity and phase velocity from ambient noise cross-correlations. *Journal of Geophysical Research: Solid Earth*, *123*(11), 9933-9949.

Reviewer: 1

SUMMARY

This is an impactful paper, in terms of methodological developments and also a societally relevant application. The method (ambient noise differential adjoint tomography) is a new variant on two methods that are relatively new to seismology. Ambient noise adjoint seismology was first performed by Min Chen et al. (2014 GRL: Low wave speed zones in the crust beneath SE Tibet revealed by ambient noise adjoint tomography), while travelttime differential adjoint tomography has been performed with P waves by J. Chen (2022) and P. Chen (2023). Here the authors blend these approaches and aim the technique at a compelling target: shallow, groundwater-bearing rocks in the Los Angeles region. (The image spans a linear profile and is therefore 2D.) To me, Figure 3 (top) is the key result, as it clearly shows a contrast in elastic properties across the fault; the authors make a compelling case that this is due to difference in rock properties and groundwater reserves. The authors have done due diligence in presenting their methods as well as their caveats. I have some minor suggestions that can likely be addressed with minor updates.

Thank you very much for the constructive comments.

MAIN POINTS

+ There are some key recent papers whose work should be considered and possibly cited. I think that these two papers are the first applications of differential adjoint tomography:

Chen, J., Chen, G., Wu, H., Yao, J., & Tong, P. (2022). Adjoint tomography of northeast Japan revealed by common-source double-difference travel-time data. *Seismological Research Letters*, 93(3), 1835–1851. <https://doi.org/10.1785/0220210317>

Chen, G., Chen, J., Tape, C., Wu, H., & Tong, P. (2023). Double-difference adjoint tomography of the crust and uppermost mantle beneath Alaska. *Journal of Geophysical Research: Solid Earth*, 128, e2022JB025168. <https://doi.org/10.1029/2022JB025168>

The distinction is that they used P waves, whereas the current study uses ambient noise surface waves. But the fact that the principles seem to work in a real problem was demonstrated with P waves.

A paper using adjoint tomography (or "full waveform inversion") to examine groundwater reservoirs is this one:

Wang, W., Nyblade, A., Mount, G., Moon, S., Chen, P., Accardo, N., et al. (2021). 3D seismic anatomy of a watershed reveals climate-topography coupling that drives water flowpaths and bedrock weathering. *Journal of Geophysical Research: Earth Surface*, 126, e2021JF006281. <https://doi.org/10.1029/2021JF006281>

<https://agupubs.onlinelibrary.wiley.com/doi/full/10.1029/2021JF006281>

It's possible that the scales and techniques are quite different, but it may be worth citing.

Thanks. We add the references in the text at lines 318-322 and 329-330.

“Double-difference adjoint tomography using P wave has been applied to northeast Japan (Chen et al 2022) and Alaska (Chen et al 2023) for crust and upper mantle structure. For earthquake body wave

data, this method is capable of canceling the signature of earthquake source radiation pattern, which is helpful in enhancing the resolution of the tomographic image.”

“The ambient noise differential adjoint tomography has a similar goal for canceling the noise source signature.”

At lines 49-51:

“The former method is quite expensive and logistically difficult in an urban area, but can provide information on shallow water flowpath based on P wave velocity (Wang et al 2021)”

+ The theory to estimate porosity directly (rather than indirectly, as done here) is fully established (Morency, Tromp, 2008, 2009, 2011) and not yet applied for 3D real problems, as far as I know.

I think that Andrea Revil also tackled poroelastic wave propagation and seismic imaging:

<https://scholar.google.com/citations?user=dsQqrkwAAAAJ&hl=en>

Thanks. We also add references on the theory of porosity estimation with adjoint tomography at lines 147-148:

“ One theoretical framework could estimate directly the porosity information using adjoint state method, but it has no practical application to date.”

+ I wonder if the V_p/V_s contrast across the fault could be demonstrated more clearly by comparing a simple depth profile of V_p/V_s for the average of one side and the average of the other side.

Potentially this could be included in Figure 3.

Thanks. We compute the average V_p/V_s for $X=0\sim 5\text{km}$ and $X=5\sim 10\text{km}$, respectively. The figure is attached below. We decide not to include this figure in the manuscript as the information is the same as Figure 3(top).

Figure R1. Averaged V_p/V_s ratio curves for the western and eastern blocks around Newport-Inglewood Fault.

MISC POINTS

+ Put and A and A' in Figure 3 or just NE and SW at the ends of the top image.

Thanks. We add the labels *A-B* in Figure 3 and Figure 1.

+ I did not see any mention of what method was used to solve the 2D elastic wave equation (finite difference? spectral element?). This should be stated in the Methods.

Thanks. We use the finite difference method. We add this information in Method section at lines 398-399:

“We use finite-difference method to compute the elastic wavefield for forward and backward simulations.”

+ Is topography taken into account? It is shown in Figure 2a, but is it accounted for in the forward wavefield simulations?

Thanks. Our current forward wavefield simulation code does not handle topography. It assumes a flat surface at the top. However, we mainly study the sedimentary layers for *X* between 0 and 26 km, where the highest topography change (~30 to 40 m) occurs at the Newport-Inglewood Fault. Therefore, topography should not be a major concern for this study.

+ Is the numerical grid regular or are there smaller elements near the surface in order to efficiently accommodate the lower wave speeds? Certain considerations would be essential for extension to 3D.

Thanks. We only use a regular grid and finite-difference method for this work. We will try the adaptive meshing for future 3D applications of differential adjoint tomography.

+ Rayleigh wave ellipticity measurements, such as those used in Berg et al. (2018 JGR), have the potential to support very shallow seismic imaging. These can be extracted from either earthquake waves or ambient noise.

Thanks. We also add references to Berg et al. (2018 JGR) in the discussion section:

“Using the regional broadband network in Southern California, a more recent study combines the phase velocity and ellipticity information from ambient noise Rayleigh wave to estimate the near surface S wave velocity model (Berg et al. 2018), which improve upon the SCEC community velocity model.”

+ Castellanos2020 is cited for the V_p model. It seems like there is potential to use a similar approach to get V_p as was used to get V_s . For example, could the P waves from Castellanos2020 or from earthquakes be used within a differential adjoint tomography framework? This would ensure that the 3D wavefield simulations were directed at both V_s and V_p . The authors address this topic in the final paragraph of the paper, and I'm not suggesting that more needs to be stated.

Thanks. We appreciate the idea to suggest using differential adjoint tomography for V_p model derived in Castellanos2020. We also add this suggestion in the discussion section at lines 306-308:

“For future work, it would be interesting to implement double-difference adjoint tomography for the same P wave travel time data from Long Beach array, which can potentially improve the P wave velocity tomography result.”

Reviewer #2

This study focuses on a new adjoint tomography method which improves existing imaging methodologies in two important ways: (1) produces better images of the shallow seismic structure using ambient noise, and (2) minimizes the bias from uneven noise source distribution which often leads to inaccurate velocity estimations. The authors applied this method to a field case study, demonstrating its feasibility. Generally speaking, I think this study is worthy of publication, but would need a major revision of the physical connection between the seismic observations and their geophysical significance/importance to the LA Basin/NIF. I'd like to point out the following comments/questions:

Thank you very much for the constructive comments.

Adjoint Method

1. although the authors have addressed the advantage of this new method compared to the conventional ray-theory inversion method (i.e. lines 51-58), it is still a little unclear to me how much increased sensitivity is expected. I assume this new method can do a better job to image the seismic velocity heterogeneity (because this method is able to identify a shallow aquifer which was not seen from other tomography studies). Better highlighting this would enhance the paper.

Thanks. To highlight the advantage of the new adjoint tomography method in imaging a shallow aquifer, we add a few sentences to lines 68-70.

“Our differential adjoint tomography successfully images a shallow aquifer between 0.2-0.4 km depth and other deeper fluid-bearing structures in the top 2 km depth due to the enhanced sensitivity to S wave velocity structure.”

2. It might be good to highlight the importance of studying near-surface seismic structure. why should we care about the near-surface seismic structure, and what are the challenges in doing shallow structure imaging using the existing techniques?

Thanks. We add the following sentences highlighting the importance at lines 43-53:

“The near-surface structure in urban areas is important in several aspects. It supports the weight of building foundations, and the near-surface velocity/attenuation structure is important for ground motion prediction of potential earthquakes. It also hosts aquifers that could play an important role in the water supply, and the spatial distribution of the aquifers could also affect the subsidence of construction or liquefaction in times of an earthquake. Traditional methods for exploring the near-surface structure include seismic surveys with artificial sources (explosion or vibroseis), and electrical resistivity. The former method is quite expensive and logistically difficult in an urban area, but can provide information on shallow water flowpath based on P wave velocity. The latter approach is more sensitive to the very shallow sediments in the top 50 m depth and has limited spatial coverage of ~300 m.”

3. It might be better to place one sentence to introduce what LASSIE array is (i.e., what kind of data is it recording? line 50)

Thanks. We add one sentence at lines 63-64:

“The LASSIE array consists of 42 broadband seismic stations that record the seismic noise data for one month in 2014.”

Vp/Vs – Porosity Connection and Geophysical Interpretation

1. The derived porosity based on vp/vs is plausible. I wonder if there is any other evidence/data/reference to support the derived porosity value. Some follow-on questions that might help you answer the above question, but don't necessarily need to be addressed specifically – perhaps a geologic cross section exists along the same cross section of the NIF as Figure 3 and could show some geologic heterogeneity? Is the San Pedro Formation not present on the east side of the fault? How much throw is on the NIF? Is that coincident with your Vp/Vs gradient across the fault?

Thanks. The LBCC well is the only water well with core samples tied to the sequence stratigraphy near the first 5km of the SW LASSIE array (e.g. Ponti et al 2007). We could not find another borehole with core stratigraphy in the top 500 m near the LASSIE array segment between NIF and LAF (X=5~10 km). The observed Vp/Vs contrast in the top 2 km across the NIF (Fig.3) is more likely due to the bimaterial interface of NIF, which separates apart the two blocks with strike-slip motion and therefore the physical properties (velocity, porosity, etc.) are unlikely to be the same.

Based on the structure map of Long Beach near Signal Hill (reference below) where the LASSIE array crosses, the throw across NIF is ~ 700 ft, or ~ 210 m for the eastern block. The 210-m uplift of the eastern block probably brought the water-bearing structure to a shallower depth. Therefore, we cannot identify the Silverado aquifer between 0.2-0.4 km depth in the eastern block (X=5~10 km).

We clarify this point between lines 142-145:

“Across the Newport-Inglewood Fault, there is a 210-m uplift of the eastern block according to the structure map beneath Long Beach near Signal Hill, which probably brought the water-bearing structure to a shallower depth. Therefore, we cannot identify the Silverado aquifer between 0.2-0.4 km depth in the eastern block”

Division of Mines & Geology. (1940). *Geologic Formations and Economic Development of the Oil and Gas Fields of California*. California State Printing Office.

<https://books.google.com.hk/books?id=9j9bQfD7YzYC>

2. The words "porosity" and "permeability or permeable" should not be mixed (i.e., line 140). High porosity rock does not always lead to high permeability. Clays are an example of a rock type that exhibit high porosity and low permeability. I'd encourage the authors to mention the definition and address the correlation between the two in the manuscript.

Thanks. We add definitions of porosity and permeability in the text. In addition, we also mention the connections between these two terms on lines 167-169:

“ For sand, higher permeability correlates with higher porosity. For clay, its permeability is low despite higher porosity.”

While the presence of clay can constrain the flow directions due to its low permeability, the sand layer in between clay layers can form a fluid reservoir and allow fluid to flow within the sand layer. We think the Silverado aquifer consists of several sand layers that hold water and several underlying/overlying clay layers that seal the aquifer. At some horizontal locations, however, the clay layers may be too thin or contain cracks, allowing fluid to migrate downward. We add clarifications between lines 169-171.

3. The porosity predictions are quite sensitive to the assumed geology (i.e. clay vs. sand fraction). There is a difference of more than 100% in porosity. What doesn't quite make sense to me – clay's usually have much higher porosities than sands. Clay's typically have total porosities in the 40-50% range but very low effective porosity (1-2%) due to the plate-like arrangement of grains. This very low effective porosity gives clays their extremely low permeability. Sands typically exhibit total porosities in the 20-25% range, but have orders of magnitude higher permeabilities as the spherical grains produce effective porosities near the total porosity. Is the BGTL model actually sensitive to effective porosity and not total porosity? Effective porosity is more directly related to permeability...

Thanks. The BGTL model is sensitive to the total porosity. The method alone cannot differentiate between clay or sand. For the same V_p/V_s ratio, the BGTL method will predict a much lower porosity for clay due to its much smaller shear modulus compared with sand. In Fig3, the predicted porosity is too low for clay and too high for sand, suggesting that either clay or sand alone cannot explain the observed V_p/V_s ratio. So we choose a mixture model of clay and sand, in which the clay with higher porosity and sand with lower porosity coexist to yield a reasonable porosity estimation.

As we don't know the exact fractions of clay & sand, we assume half clay and half sand for porosity estimation. This model is not for exact prediction of porosity, but rather for estimation of lateral variation in porosity/pore fluid fraction as related to S wave velocity.

We clarify these points between lines 158-161:

“For the same V_p/V_s ratio, the BGTL method will predict a much lower porosity for clay due to its much smaller shear modulus compared with sand. Therefore, the predicted porosity is too low for clay and too high for sand, suggesting that either clay or sand alone cannot explain the observed V_p/V_s ratio.”

4. Line 220-221, the authors argue the seismicity around the Newport-Inglewood fault could be related to high amount of pore fluids suggested by high porosity (~ 0.33). Higher porosity values, in and of themselves, do not mean that fluid pressure is higher or lower in this portion of the aquifer. Porosity influences aquifer storage, but not necessarily fluid pressure. Higher porosity rocks tend to be higher permeability (with some exceptions like those noted above), but higher permeability would actually tend to indicate lower fluid pressures are hydraulic gradients would not build as easily. We expect the groundwater flow to occur more easily without causing high pore pressure to accumulate, so the fault can stay strong (e.g., Townend and Zoback, 2000).

Thanks. High porosity rocks in the Silverado aquifer may be corresponding to high permeability and the groundwater can flow within the aquifer, so they may not affect the effective normal stress on the NIF fault (e.g., Townend and Zoback, 2000). But the deeper fluid reservoir between 0.8-2.2 km depth around the NIF could be sealed as it corresponds to a known oil field around NIF beneath the Signal Hill. In that case, the pore fluid can reduce the effective normal stress and cause micro seismicity.

We add clarifications between lines 257-260:

“Although the fluid within the Silverado aquifer flows easily and cannot increase the pore pressure (e.g., Townend and Zoback, 2000), the deeper fluid reservoir between 0.8-2.2 km depth may be sealed and can increase the pore fluid pressure due to the combination of hydrocarbon reservoir beneath Signal Hill and water injection.”

Division of Mines & Geology. (1940). *Geologic Formations and Economic Development of the Oil and Gas Fields of California*. California State Printing Office.
<https://books.google.com.hk/books?id=9j9bQfD7YZYC>

Grammatical Edits

1. Line 43: lower Pleistocene age

Thanks. Fixed.

2. Line 46: deeper aquifers? Water is likely present at deeper depths but perhaps not aquifer quality/quantity

Thanks. Fixed.

3. Line 166: check for grammatical clarity

Thanks. Fixed.

4. Line 182: check for grammatical clarity

Thanks. Fixed.

Reviewer #3

Summary:

The authors propose an ambient noise differential adjoint tomography method, apply it to the LASSIE array, and further come out with a shear wave velocity model beneath the LA basin. Owing to the advantages of using differential time information between receivers, they eliminate the potential bias caused by unknown non-uniform noise sources. More than a velocity model, they also estimate V_p/V_s ratio and porosity according to some previous works. They found extremely high V_p/V_s ratios at the place where the Silverado aquifer was observed and indicating the existing pore fluids.

Thank you very much for the constructive comments.

Comments:

1. Data information needs to be included somewhere in the manuscript.

Thanks. We include the data information for LASSIE array in Data availability section.

2. In the synthetic results (Figure 5), the main feature was revealed, but there are many oscillations at sidelobes. As well as Figure 6, many oscillations exist in the kernel profiles. What is the cause of these oscillations? Are these patches appear due to finite frequency bands? For the V_s models

(Figure 2), most variations are in shallow portions, and they look like the oscillations in the synthetic tests. Can the authors clarify these small patches of anomalies from the oscillations as the synthetics? The fluctuations appear in Figure 3 as well at ~1 km to the surface.

Thanks. The oscillations are sidelobes of the banana-doughnut kernel, which arise from the Born approximation used in deriving the Fréchet derivative (kernel) of misfit function (e.g. Tromp et al., 2004) and they are common for adjoint tomography. They also arise from interference effects between different parts of the wavefront used in kernel computations. And they are related to the finite-frequency bands as one needs to specify the frequency of the source-time function when computing the kernel. These kind of oscillations are more obvious in Figure 6A&B, where we compute the differential-time kernel for one station triplet. But the amplitude of the side lobes are much smaller than the main lobe. Therefore, when we combine all the station triplets and virtual sources, we can suppress most of these oscillations (Figure 7A). The shallow features in the Vs model (Figure 2 & 3) are real and related to the combined kernel in Figure 7A&B. The only remaining effect of side lobes would appear at the edges of an array, but the side lobes are still much weaker than the main lobes, otherwise the misfit functions for training and validation sets would not decrease.

We add explanations of Born approximation & side lobes at lines 407-412:

“For both station triplets, there are asymmetric side lobes to both sides of the differential-time receiver pair. The blue side lobe next to the left receiver (closer to virtual source) is possibly caused by differential time measurements, finite-frequency effect and the Born approximation used in computing the Fréchet derivative of the misfit function, but the amplitudes of the side lobes are much smaller than the main lobe.”

In addition, we explain the cancellation of side lobes at lines 419-423:

“When we combine all the virtual sources with possible station triplets, we can suppress most of these oscillations (Figure 7A). The only remaining effect of side lobes would appear at the edges of an array, but the side lobes are still much weaker than the main lobes, otherwise the misfit functions (Figure 7) for training and validation sets would not decrease.”

3. Line 122-124. This sentence may not be appropriate. The purpose and the scales of this study and the cited paper (Shaw et al., 2015) are different. Why do the authors want to mention Shaw et al. (2015) as a point? Moreover, there are many other Vs models that have been published (e.g., Lin et al., 2013; Berg et al., 2018, etc.) but have yet to be mentioned in the manuscripts. The authors may want to add paragraphs to demonstrate the comparison between their Vs model and other published Vs models.

Thanks. We agree that the purpose and scales of this study is different from Shaw et al (2015), which is part of the CVM-H model used in SCEC.

We have added the following related paper references on Vs models.

Lin et al. (2013) use the Long Beach dense 2D array, which contains more than 5200 receivers with average spacing of 100 m. They use the Eikonal tomography method to estimate phase velocity map at different frequencies first. Then they perform 1D inversion of S wave velocity structure below each geographic location.

Berg et al. (2018) compute the Vs model using the regional CI network broadband stations in Southern California. They combine the phase velocity and ellipticity information from Rayleigh wave to estimate the near surface S wave velocity model, which provides constraints for the top 9 km structure.

We add new citations and explanations at lines 283-293:

“Using the regional broadband network in Southern California, a more recent study combines the phase velocity and ellipticity information from ambient noise Rayleigh wave to estimate the near surface S wave velocity model, which improve upon the SCEC community velocity model. Another study derives the S velocity model with the Long Beach nodal array, which contains more than 5200 receivers with average spacing of 0.1 km, while the LASSIE array has average spacing of 1 km. They use the Eikonal tomography method to estimate phase velocity map at different frequencies, which takes the spatial gradient of travel time for each virtual source. Unlike the adjoint tomography that directly estimate S velocity, the Eikonal tomography converts phase velocity maps to S velocity structure through 1D depth inversion below each geographic location. “

We also add comparison between our Vs with Lin et al. (2013) at lines 111-114:

“A previous study using Long Beach array also finds slower S velocity perturbation between 0.1-0.4 km depth near the southwest end of the LASSIE array (Lin et al. 2013). But their velocities at this depth range are generally more than 35% faster than results shown here at 300-m depth.”

4. What is the spatial resolution for the Vs model (this research) and the used Vp model (the referred article) in the Long Beach section? The nodal array for the Vp model (Castellanos et al., 2020) looks much denser than the LASSIE array, but the Vp profile (Figure S2) looks like a reference 1-D model. As mentioned in many places of the manuscript, Vp and Vs models are from different methods and have different sensitivity to the structures. It would be good to discuss how these two models are sensitive to the structures differently. As said in Lines 244-252, there may be existing some unresolved velocity anomalies.

Thanks. The lateral spatial resolution of our study is limited by the LASSIE array station spacing, which is ~1.0 km. The Vp model in the Long Beach section is derived using P wave travel time and ray tracing technique (Castellanos et al., 2020), and the 2D Long Beach array has a dense spacing of ~0.1 km. The P wave ray path between a virtual source and a receiver turns at approximately 2 km depth and becomes almost vertical near the surface, which is not sensitive to the shallow, thin aquifer system due to the nearly vertical incidence. The Vs model is based on Love wave (surface wave) traveling horizontally, which is suitable for resolving lateral variations of Vs in the shallow aquifer system.

We add explanations at lines 297-302:

“The P wave ray path between a virtual source and a receiver turns at approximately 2 km depth and becomes almost vertical near the surface, which is not sensitive to the shallow, thin aquifer system due to the nearly vertical incidence. The Vs model derived here is from the ambient noise differential adjoint tomography based on wave equations and a linear array, which involves surface wave traveling horizontally and is suitable for resolving lateral variations of Vs in the shallow aquifer system.”

5. To study the structures at Long Beach, there is a comparable Vs model using the same nodal array as the Vp model proposed by Lin et al. (2013). What is the resolution difference between the Vs model in this study and Lin’s model?

Thanks. Our study and Lin et al. (2013) are both based on ambient noise surface waves. The LASSIE array has lateral resolution ~1.0 km as the average station spacing, while Lin et al. (2013) has a lateral resolution of ~ 0.1 km as the station spacing in Long Beach dense array.

We add related explanations at lines 287-293, next to the reference to Lin et al (2013) in discussion.

6. In Figure 3 black curve (and Line 136), the local minimum porosity ($X \sim 1.6$ km) has a bit shift to $X = 2$ km (where the highest V_p/V_s ratio appears at the top plot). Can the authors clarify this?

Thanks. The black curve shows the local minimum porosity at $X \sim 1.6$ km, which is probably related to the variable thickness of clay layer above 0.8 km depth that control the how much fluid can migrate downwards. There is also a gap in station coverage between $X = 1 \sim 2$ km, which might shift the local porosity minimum towards $X = 1.6$ km.

7. The manuscript mainly discusses the Long Beach section's V_p/V_s ratios and porosity. How about the rest of the areas in the profiles? For instance, there is a velocity reversal at a distance between 25-35 km in Figure 2.

Thanks. As we don't have V_p model outside Long Beach, we can only rely on the V_s model for interpretations of the rest of areas. For X between 26-35 km, there is a quite a bit of topography ~ 300 m and our current code does not handle topography. So we do not interpret the results for $X = 25 \sim 35$ km. For Figure 2C, the area to the east of LAF ($X = 10 \sim 13$ km; depth between 0.2 \sim 1.8 km) also corresponds to significant V_s reduction of ~ 20 -30%, suggesting possible presence of pore fluids.

We add related explanations at lines 237-241:

“For example, the area to the east of LAF ($X = 10 \sim 13$ km; depth between 0.2 \sim 1.8 km) also corresponds to significant V_s reduction of 20 \sim 30%. The shallow sections (0.1 \sim 0.5 km depth) between NF and LEPTF or at the western foothill ($X = 25$ km) also feature significant V_s reduction 20 \sim 25%.”

8. While the topography along the LASSIE array does not change much in the WS part, where the paper mainly focuses, do the authors perform/consider elevation correction in the resulting model since they target shallow top 1 km depth?

Thanks. Our current code does not handle topography. It assumes a flat surface at the top. However, we mainly study the sedimentary layers for X between 0 and 26 km, where the highest topography change (~ 30 to 40 m) occurs at the Newport-Inglewood Fault. Therefore, topography should not be a major concern for this study.

9. In the last part of the main text (Lines 356-380 and Fig7), the authors demonstrate how the iterative process goes throughout the iteration. The results shown in this section are based on the low-frequency band (0.16-0.35 Hz). How does the iterative process go when using low+high frequency signals (as shown in Figure 2C)?

Thanks. We add a figure to the Supplementary Material (Figure S3) for the iterative process of low+high frequency signals. The figure is also attached below:

Figure R2. Misfit function versus iterations for low+high frequency signals.

10. Using a small dataset to validate the output model is common when resolving scientific problems. This concept is not originally from the machine learning or deep-learning community. It seems unnecessary to mention deep-learning neural networks (Lines 61-62) or machine learning (Lines 240-241).

Thanks. We borrow some concepts such as training and validation sets from machine learning. So the references in the text are just analogies of some concepts with machine learning. We prefer to keep those references as a reminder of the overfitting issue in adjoint tomography.

11. What is the unit of misfit? The values of the misfit in Figure 7 look large.

Thanks. The unit is in sec^2 . It seems large because it is a sum of the squared differential time misfit over all frequencies and all possible station triplet combinations.

12. It would be good to mention Fig 2C in the paragraph (Lines 100-105).

Thanks. We add reference to Fig 2C in this paragraph.

13. Lines 169-171: What do the authors mean that both Vs100 and Vs30 “decrease” at the edge of the unit? Consider marking the faults in the bottom plot of Figure 4.

Thanks. The Vs30 was derived from sparse samples, so they divide the LA basin into different geologic units and assign constant or slope value to each unit. We have marked the faults in the bottom plot of Figure 4. We also modified this sentence:

“For the geological unit between the Los Alamitos Fault and the Lower Elysian Park Thrust Fault, both the inverted Vs 100 (adjoint tomography) and the Vs 30 decrease at the edges of the geologic unit defined in Vs 30 model.”

14. How do the five virtual source stations be selected?

Thanks. For the validation set, we choose 3 virtual sources at the SW side and 2 virtual sources at the NE side of the array. We expect it to be a balanced choice and ensures that we can find overfitting

issues for the structure between $X=0\sim 25$ km.

For figures:

- Figure 1: (1) The solid triangles and the red dot (the well location) are hard to read. (2) consider marking the section of Figure 3.

Thanks. We change the color of red dot and solid triangles. We also mark the section A-B of Figure 3.

- Figure 4: Consider marking the faults in the bottom plots.

Adding color-code units (or components) on those figures with color bars would be better.

Thanks. We add faults in the bottom plots of Figure 4. We also add units to the color bar.

Reviewer #2

(Remarks to the Author)